# Genetic Variants Associated with Drug Resistance of Cytomegalovirus in Hematopoietic Cell Transplantation Recipients

**DOI:** 10.3390/v15061286

**Published:** 2023-05-30

**Authors:** Seungwan Chae, Hoon Seok Kim, Sung-Yeon Cho, Dukhee Nho, Raeseok Lee, Dong-Gun Lee, Myungshin Kim, Yonggoo Kim

**Affiliations:** 1Department of Laboratory Medicine, Seoul St. Mary’s Hospital, College of Medicine, The Catholic University of Korea, Seoul 06591, Republic of Korea; cswan0817@naver.com (S.C.); hskim11@catholic.ac.kr (H.S.K.); 2Catholic Genetic Laboratory Center, Seoul St. Mary’s Hospital, College of Medicine, The Catholic University of Korea, Seoul 06591, Republic of Korea; 3Division of Infectious Diseases, Department of Internal Medicine, College of Medicine, The Catholic University of Korea, Seoul 06591, Republic of Korea; cho.sy@catholic.ac.kr (S.-Y.C.); nhodh@catholic.ac.kr (D.N.); misozium03@catholic.ac.kr (R.L.); symonlee@catholic.ac.kr (D.-G.L.); 4Vaccine Bio Research Institute, College of Medicine, The Catholic University of Korea, Seoul 06591, Republic of Korea; 5Catholic Hematology Hospital, Seoul St. Mary’s Hospital, Seoul 06591, Republic of Korea

**Keywords:** cytomegalovirus, drug resistance, viral, pharmacogenomic variants, hematopoietic cell transplantation, *UL97*, *UL54*

## Abstract

Cytomegalovirus (CMV) infection is a serious complication in hematopoietic cell transplantation (HCT) recipients. Drug-resistant strains make it more challenging to treat CMV infection. This study aimed to identify variants associated with CMV drug resistance in HCT recipients and assess their clinical significance. A total of 123 patients with refractory CMV DNAemia out of 2271 HCT patients at the Catholic Hematology Hospital between April 2016 and November 2021 were analyzed, which accounted for 8.6% of the 1428 patients who received pre-emptive therapy. Real-time PCR was used to monitor CMV infection. Direct sequencing was performed to identify drug-resistant variants in *UL97* and *UL54*. Resistance variants were found in 10 (8.1%) patients, and variants of uncertain significance (VUS) were found in 48 (39.0%) patients. Patients with resistance variants had a significantly higher peak CMV viral load than those without (*p* = 0.015). Patients with any variants had a higher risk of severe graft-versus-host disease and lower one-year survival rates than those without (*p* = 0.003 and *p* = 0.044, respectively). Interestingly, the presence of variants reduced the rate of CMV clearance, particularly in patients who did not modify their initial antiviral regimen. However, it had no apparent impact on individuals whose antiviral regimens were changed due to refractoriness. This study highlights the importance of identifying genetic variants associated with CMV drug resistance in HCT recipients for providing appropriate antiviral treatment and predicting patient outcomes.

## 1. Introduction

Cytomegalovirus (CMV) infections are usually asymptomatic and self-limiting in healthy adults. However, they can cause life-threatening infections with high mortality in immunocompromised patients such as neonatal infants, patients with immunodeficiency diseases, cancer patients, and recipients of hematopoietic cell transplantation (HCT) [1,2,3,4]. CMV infection manifests as viremia within the circulatory system. As the disease progresses, it triggers inflammation in a variety of organs, including but not limited to the gastrointestinal tract, retina, lungs, and meninges, ultimately resulting in patient death [5,6,7]. In recipients of HCT, CMV infection can exert direct pathogenic effects on multiple organs and indirect effects such as graft-versus-host disease (GVHD), hematopoietic cell graft failure, and concurrent diseases with other pathogens [7,8,9]. Ultimately, CMV infection is linked to poor clinical outcomes, including a higher risk of overall death and non-relapse mortality after HCT [10,11]. Clinicians tasked with the care of HCT recipients must be vigilant for CMV infection. This can be achieved through pre-transplant screening for CMV serostatus and routine monitoring of the CMV viral load. Pre-emptive and prophylactic antiviral therapies such as ganciclovir, valganciclovir, foscarnet, cidofovir, and letermovir should be implemented to manage CMV infection in HCT recipients [12,13].

Extended and repetitive use of antiviral agents for the management of CMV infection has resulted in the appearance of drug-resistant viral strains [14,15]. Genetic variations associated with resistance to antiviral agents have been identified primarily in viral kinase (*UL97*) and DNA polymerase (*UL54*) genes [16,17]. Upon acquiring genetic variants, CMV may acquire resistance to corresponding antiviral agents, potentially leading to multidrug resistance [18,19]. Therefore, early identification of genetic variants associated with drug resistance is essential not only to facilitate individualized modifications of antiviral therapy for non-responsive patients but also to improve patient outcomes [20]. This study aimed to identify genetic variants associated with drug resistance in the *UL97* and *UL54* of CMVs that have been responsible for refractory DNAemia among HCT recipients. This study also aimed to evaluate the clinical significance of these variants.

## 2. Materials and Methods

### 2.1. Patients

A total of 2271 patients who received HCT at the Catholic Hematology Hospital between April 2016 and November 2021 were included in this study (Figure 1). Among them, 1428 patients (62.9%) received antiviral therapy with CMV DNAemia above the threshold. Due to refractory CMV DNAemia, 123 (8.6%) of them also had DNA sequencing requested to identify drug resistance variants. The medical records of those 123 patients were carefully reviewed to look into their demographic characteristics, diagnosis, laboratory data (including the donor’s and patient’s CMV serostatus), treatment, and clinical outcome. These clinical data were used to evaluate the clinical relevance of the genetic variations found in the initial CMV drug-resistant test results. Due to the multiple requests from the same patient at different times, a total of 160 CMV DNA samples were analyzed for CMV drug resistance testing (Figure 1). Among 24 patients who underwent multiple CMV drug resistance testing, 14 patients were tested twice, 7 patients were tested 3 times, and 3 patients were tested 4 times. This study was reviewed and approved by the Institutional Review Board (IRB) of Seoul St. Mary’s Hospital (KC22RASI0817). The requirement of informed consent was waived by the IRB due to its retrospective study design.

### 2.2. Monitoring of CMV Infection

Prior to HCT, CMV serostatus was analyzed according to the results of CMV IgG using an IMMULITE 2000 XPi Immunoassay System (Siemens Healthineers, Erlangen, Germany). After HCT, the plasma CMV DNA level was regularly checked. Depending on the post-transplantation date, monitoring was done twice a week for day 0–30, once a week for day 30–100, and once every two weeks from day 100 onward.

DNA was extracted from EDTA plasma using QIAsymphony SP (QIAGEN, Hilden, Germany) with a QIAsymphony DSP DNA Mini Kit (QIAGEN). The CMV viral load was measured by real-time quantitative PCR test using an artus CMV QS-RGQ MDx Kit (QIAGEN) and a Rotor-Gene Q (QIAGEN) according to the manufacturer’s instruction. The copy number of CMV DNA was based on the standard curve using CMV standard DNA. It was converted to fit the volume to obtain copies per milliliter. Results (IU/mL) are reported together with a conversion factor of 1.64. The limit of detection was 69.7 IU/mL. The peak CMV viral load was defined as the highest measured value during CMV viral load monitoring in a patient.

### 2.3. Treatment Strategies for CMV Infection

During regular observation, pre-emptive therapy with ganciclovir, valganciclovir, or foscarnet was initiated when the CMV viral load exceeded 500 IU/mL in the high-risk group and 1000 IU/mL in the low-risk group [21]. Refractory CMV DNAemia was suspected if the CMV viral load did not decrease by more than 1 log_10_ after two weeks of pre-emptive therapy [6,22,23]. In patients with suspected refractory CMV DNAemia, we decided to change the antiviral agent after consulting with infectious disease physicians and performed a CMV drug resistance test. In addition, we administered letermovir as an anti-CMV prophylaxis to CMV seropositive patients who underwent allogeneic HCT after September 2020 [13].

### 2.4. CMV Drug Resistant Testing

To perform CMV drug resistance testing, we employed direct sequencing for *UL97* and *UL54* genes. To analyze the *UL97* gene, forward and reverse primers (HLF97_F; 5′-CTG CTG CAC AAC GTC ACG GTA CAT C-3′ and HLF97_R; 5′-CTC CTC ATC GTC GTC GTA GTC C-3′) were used. To analyze the *UL54* gene, 3 pairs of forward and reverse primers (UL54-1_F; 5′-GAG TTC CCT TCC GAA TAC GA-3′, UL54-1_R; 5′-AGC GTT AGG TGA CAC AGC AA-3′, UL54-2_F; 5′-GTA TTG GTG CGC GAT CTG TT-3′, UL54-2_R; 5′-CCA CGG GGT CGT TGT AGT AA-3′, UL54-3_F; 5′-GCG TTT CCA ACG ACA ATC AC-3′ and UL54-3_R; 5′-CGT GCG CTC TAG CAT GTC-3′) were used. PCR was performed using a C1000 Touch Thermal Cycler (Bio-Rad, Hercules, CA, USA). Amplified DNA was sequenced using an Applied Biosystems BigDye Terminator v3.1 Cycle Sequencing Kit (Thermo Fisher Scientific, Waltham, MA, USA) and an Applied Biosystems 3500xl Dx Genetic Analyser (Thermo Fisher Scientific). The sequencing PCR process was initiated with an initial denaturation step at 96 °C for 1 min, constituting 1 cycle. Subsequently, the examination proceeded with the following conditions; denaturation at 96 °C for 10 s, annealing at 50 °C for 5 s, and extension at 60 °C for 4 min. This process was repeated for an additional 25 cycles. Finally, the examination was completed by holding the reaction at 4 °C. The presence of variants was determined using Sequencher software (Gene Codes Corporation, Ann Arbor, MI, USA) compared with the reference strain Merlin (GenBank accession number AY446894). Detected variants of *UL97* and *UL54* genes were interpreted using the MRA Mutation Resistance Analyzer (Ulm University, Ulm, Germany) [24]. The web-based search tool analyzed variants using information stored in the database linked to published references. Through this, the variant was classified as a resistance variant, polymorphism, or variant not in the database. In our institution, ‘variant not in database’ and ‘variant with unclear phenotype’ were both defined as variant of uncertain significance (VUS).

### 2.5. Statistics

A Chi-square test was used to compare categorical variables and a one-way analysis of variance (ANOVA) test was used to compare continuous variables. Hazard ratios (HR) were reported with corresponding 95% confidence intervals (CI). The one-year survival rate was analyzed using a Kaplan Meier Estimator. All statistical analyses were performed using SPSS software ver. 24.0 (IBM, Armonk, NY, USA). A *p*-value of less than 0.05 was considered statistically significant.

## 3. Results

### 3.1. Patient Characteristics

The characteristics of the 123 patients with refractory CMV DNAemia are summarized in Table 1. Their median age was 44 years (range, 5 to 72 years). The male to female ratio was 1:1.2 (55 males and 68 females). Primary diseases observed in most patients were acute myeloid leukemia (n = 50), acute lymphoblastic leukemia (n = 24), myelodysplastic syndrome (n = 14), non-Hodgkin’s lymphoma (n = 11), and aplastic anemia (n = 9). The majority (n = 106, 86.2%) of patients underwent peripheral blood HCT, whereas 17 received cord blood transplantation. In total, 74 (60.2%) patients received HCT from related donors and 49 (39.8%) received HCT from unrelated donors. A total of 62 (50.4%) patients received HLA-matched HCT and 61 (49.6%) received HLA-mismatched HCT. A total of 13 (10.6%) patients received anti-CMV prophylaxis with letermovir around HCT. CMV seropositive rates in donors and recipients were 72.9% (70/96) and 96.7% (119/123), respectively. The median period of development of refractory CMV DNAemia was 70 days (range, 12 to 1352 days) after HCT. A total of 50 (40.7%) patients experienced CMV organ disease and 11 of these patients had the disease in more than 2 organs: colitis (n = 29), retinitis (n = 15), gastritis (n = 13), pneumonitis (n = 7), encephalitis (n = 2), and meningitis (n = 1). The peak CMV viral load was 593,178.4 IU/mL (range, 910 to 14,690,000 IU/mL), and it was significantly higher in patients with resistance variants than in others (2281,671.8 IU/mL vs. 443,754.2 IU/mL, *p* = 0.015). In total, 88 (71.5%) patients experienced moderate to severe GVHD, and the overall one-year survival rate after HCT was 52.9% (n = 65).

### 3.2. Identified Variants of UL97 and UL54 Genes

Identified variants of *UL97* and *UL54* genes are presented in Figure 2. We detected 8 different types of resistance variants in 10 (8.1%) of 123 patients (Appendix A). These variants included M460I, A594V, L595F, L595W, and C603W in the *UL97* gene and F412L, V787L, and A809V in the *UL54* gene. A594V and L595W were identified in 3 patients, and F412L and A809V were identified in 2 patients. Regarding the number of patients having a resistant variant of the two genes, 6 patients had a resistance variant of the *UL97* gene, and 2 patients had a resistance variant of the *UL54* gene. In all, 2 patients had resistance variants in both *UL97* and *UL54* genes. In addition, 1 patient showed a shift of resistance variant in the *UL97* gene from M460I to A594V.

We also detected 42 kinds of VUSs in 48 (39.0%) of 123 patients: 11 kinds of VUSs in the *UL97* gene and 31 kinds of VUSs in the *UL54* gene (Figure 2). VUSs in the *UL97* gene were located in the codon between 400 and 700, while those in the *UL54* gene were spread between codon 300 and codon 1000 [22]. In the *UL97* gene, A639T (n = 2) and R671H (n = 2) were recurrently identified in our patients. In the *UL54* gene, M827I (n = 18), T691S (n = 11), Y416H (n = 10), F460C (n = 10), and P497T (n = 10) were recurrently identified. Interestingly, our analysis revealed that Y416H, F460C, and P497T were simultaneously present in each CMV isolate. While 2 patients had a VUS of the *UL97* gene, 42 patients had a VUS in the *UL54* gene. In total, 4 patients had VUSs in both *UL97* and *UL54* genes. Of 10 patients with resistance variants, 7 also had VUSs.

Among the 24 patients who underwent multiple CMV drug resistance testing, 14 (58.3%) patients showed consistent results from all of the tests. On the other hand, 10 (41.7%) patients had changed results after repeated testing. Of these, 6 had no preexisting variants found (3 had resistance variants and 3 had VUSs). In the case of 4 patients, other variants were found (3 patients, resistance variant to VUS; 1 patient, VUS to VUS).

The frequency of genetic variation, encompassing resistance variants, VUS, and polymorphisms, was observed to be 0.857 per 100 base pairs (bp) in the *UL97* gene and 1.515 per 100 bp in the *UL54* gene. Notably, the frequency of nucleotide and amino acid changes was significantly higher in the *UL54* gene than that in the *UL97* gene (nucleotide change: 1.468 for *UL54* and 0.841 for *UL97*; amino acid change: 0.047 for *UL54* and 0.016 for *UL97*).

### 3.3. Clinical Significance of Genetic Variants

Treatment for CMV DNAemia included ganciclovir (n = 101), valganciclovir (n = 17), and foscarnet (n = 5). Patients with a peak CMV viral load above 10^5^ IU/mL (n = 70) had a lower clearance rate of the viral load than those with a peak CMV viral load below 10^5^ IU/mL (67.1% vs. 83.0%, *p* = 0.047). Of patients enrolled in this study, 44 maintained their initial antiviral regimen due to a decreasing trend in viral loads. Of these patients, 38 (86.4%) achieved CMV clearance during follow-up. Notably, patients without any resistance or VUS genetic variants exhibited a higher rate of CMV clearance than those with such variants (96.0% vs. 73.7%, *p* = 0.033, Figure 3). In total, 79 patients underwent a modification of their initial antiviral regimen due to a persistent or increasing viral load. Remarkably, the presence of any genetic variants in these patients did not significantly affect the rate of CMV clearance (66.7% vs. 67.5%, *p* > 0.05, Figure 3).

Patients with any genetic variants were more likely to have moderated to severe GVHD compared to mild GVHD (HR: 1.408, 95% CI: 1.122–1.767, *p* = 0.003, Table 1). Additionally, these patients had a poor overall one-year survival rate (HR: 1.479, 95% CI: 1.011–2.164, *p* = 0.044, Figure 4).

## 4. Discussion

In this study, we have investigated the genetic variations in the *UL97* and *UL54* genes associated with antiviral drug resistance. Additionally, we have evaluated the clinical significance of these variants in causing refractory DNAemia among HCT recipients. The overall infection rate among HCT recipients in our cohort was 62.9% (1428/2271), of whom 8.6% (123/1428) had suspected refractory CMV DNAemia. Of these individuals, 8.1% (10/123) had resistance variants upon undergoing CMV drug resistance testing. These rates are in line with previous studies reporting incidence ranges of infection rate (43.4–72.4%) and drug resistance (3.0–11.4%) [25,26,27,28].

The predominance of *UL97* gene resistance variants might have been caused by first-line agents ganciclovir and valganciclovir used for most patients (95.9% in this study) [12]. In the *UL97* gene, all resistance variants (M460I, A594V, L595F, L595W, and C603W) were located in canonical sites involved in either substrate binding or phosphate transfer of p*UL97* known to be associated with ganciclovir and valganciclovir resistance [22,24,29,30,31]. In the *UL54* gene, F412L alterations located on the catalytic site could increase the exonuclease activity. They are associated with resistance to ganciclovir, valganciclovir, and cidofovir [22,32]. V787L and A809V can alter the pyrophosphate binding site. They are associated with resistance to ganciclovir, valganciclovir, and foscarnet [14,22].

We also identified 42 VUSs in 48 patients, most of which were not present in the database. Some of these VUSs coexisted in individual CMV isolates. Of these VUSs, contrary to resistance variants, most variants were in the *UL54* gene. In the *UL54* gene, variants were distributed over large areas without canonical mutations [22]. After including polymorphisms [33,34], the frequency of genetic variations and the frequency of nucleotide and amino acid changes were higher in the *UL54* gene than in the *UL97* gene, suggesting higher susceptibility of genetic changes in the *UL54* gene than those in the *UL97* gene [35]. Several VUSs were found to coexist with other VUSs and resistance variants. While the potential impact of such coexistence on drug sensitivity remains unclear, it might have a multiplier effect on the efficacy of antiviral therapies. Thus, the precise correlation and the impact of coexisting VUSs on treatment outcomes require further investigation [25].

Our institution regularly monitored the CMV viral load in patients who underwent allogeneic HCT. If the viral load surpassed a certain level, we started pre-emptive antiviral therapy. In cases where the treatment failed to reduce the viral load, we suspected refractory CMV DNAemia and requested drug resistance testing [12]. Taking as an example the 10 patients with resistance variants identified in this study (their clinical information is shown in Appendix A), continuous follow-up revealed that 3 patients with resistance variants did not experience CMV clearance and eventually died. Even in patients with resistant mutations who experience CMV clearance, it is noticeable that CMV clearance requires a considerable amount of time. Our study revealed that patients with resistance variants had a significantly higher peak CMV viral load than those without such variants, consistent with prior research indicating a positive relationship between an elevated peak CMV viral load and the development of CMV resistance [26,27]. Patients with a higher peak CMV viral load experienced lower CMV clearance, but what preceded this outcome was unclear.

It can be challenging to identify the most effective antiviral treatment for patients who might have genetic variations associated with drug resistance. In this study, 35.8% of patients showed a decreasing trend in viral load after undergoing drug resistance testing. Notably, among these patients, the CMV clearance was dependent on the presence of genetic variants (resistance or VUS). However, in addition to that, in this study, we divided the patients with suspected refractory CMV DNAemia into two groups: those who underwent initial antiviral regimen modification and those who did not. Within each group, we further categorized patients based on the presence or absence of genetic variants detected by CMV drug resistance testing. We then assessed CMV clearance in both patient groups. Interestingly, there was no significant difference in CMV clearance rates between patients with modified antiviral regimens based on the presence of genetic variants. However, those without genetic variants in the group of patients who maintained the initial antiviral regimen had a significantly higher CMV clearance rate than those with genetic variants. This is being presented using an approach particular to this study, which has not been utilized in other studies. Moreover, patients with genetic variants exhibited a higher incidence of moderate to severe GVHD (vs. mild GVHD) and poorer overall one-year survival rate. Although we did not scrutinize each patient’s immune status, cell-mediate immunity or immunosuppressant treatment, our findings suggest that genetic information might provide useful insights into changing the antiviral regimen to overcome reduced antiviral efficacy associated with genetic variations.

The study has some limitations due to its retrospective nature, single-center design, disease heterogeneity of the cohort, and relatively small sample size of patients with resistance. These limitations may affect the generalizability and validity of the findings.

In conclusion, we sequenced drug resistant genes of CMV in HCT recipients and identified 47.2% of patients with known resistance variants and/or VUSs. It appears that these genetic variants in CMV are associated with more severe GVHD and poor outcome in patients with CMV DNAemia. Furthermore, CMV clearance was also affected by the presence of genetic variants, especially in patients who did not change their initial antiviral regimen. These findings highlight a potential association between genetic variations and reduced antiviral efficacy, implying that genetic testing might be useful for identifying patients who need alternative treatment approaches.

## Figures and Tables

**Figure 1 viruses-15-01286-f001:**
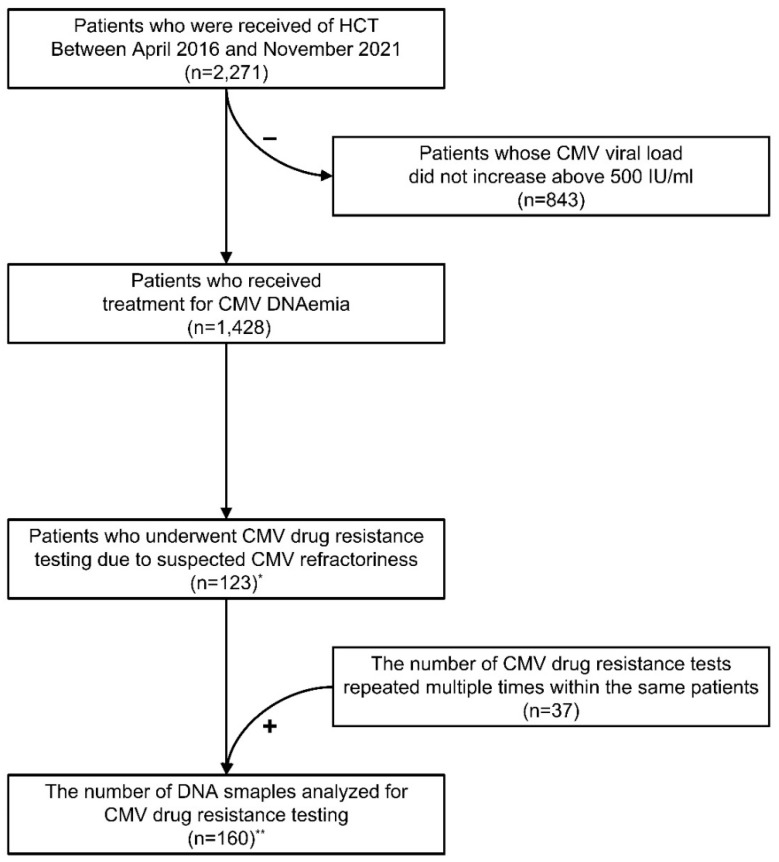
Flowchart depicting criteria used for selecting study cohorts. HCT, hematopoietic cell transplantation. * Initial CMV drug-resistant test results of the patients were used for analysis of patient characteristics and clinical significance. ** All CMV drug resistance test results, including repeated tests, were included in the genetic map of *UL97* and *UL54* genes.

**Figure 2 viruses-15-01286-f002:**
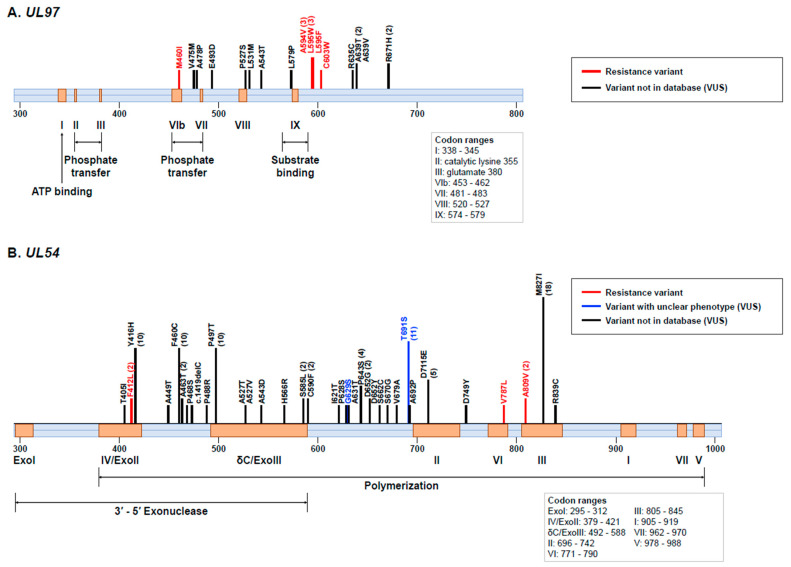
Genomic structure and map of cytomegalovirus (**A**) *UL97* and (**B**) *UL54* genes. Resistance variants and variants of uncertain significance (VUS) detected in our study are shown on the map. They are matched with functional regions in *UL97* and *UL54* genes. Variants in the *UL97* gene are found in the codon between 400 to 700, while variants in the *UL54* gene are distributed extensively. VUS includes ‘variant not in database’ and ‘variant with unclear phenotype’.

**Figure 3 viruses-15-01286-f003:**
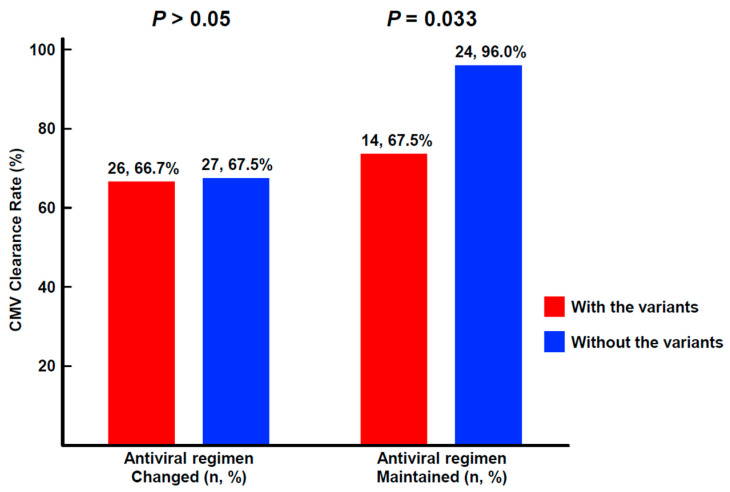
Clearance rate of cytomegalovirus viral load. Patients with their initial antiviral regimen changed showed no significant difference in cytomegalovirus (CMV) clearance (left, *p* > 0.05). Patients who continued their antiviral treatment had significantly higher CMV clearance rates in those without genetic variants than in those with variants (right, *p* = 0.033).

**Figure 4 viruses-15-01286-f004:**
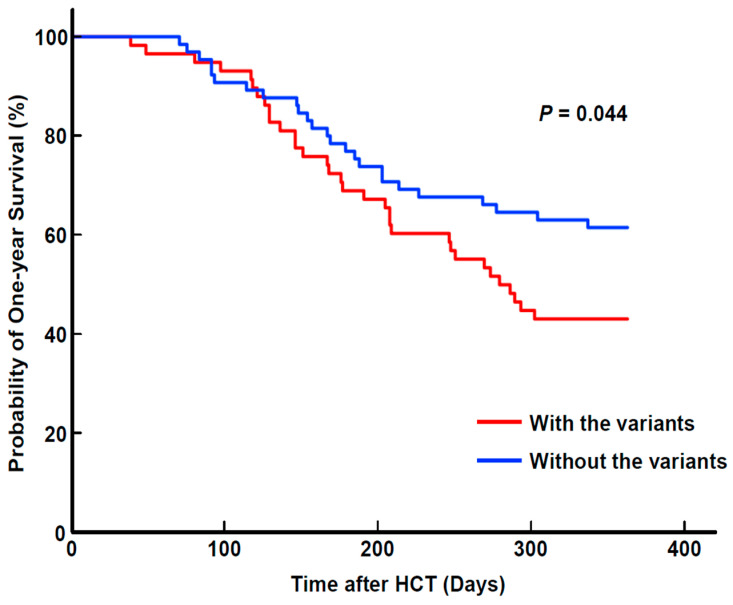
One-year survival rate after HCT in our study. The overall one-year survival rate was lower for patients with variants than for those without any variants (*p* = 0.044). HCT, hematopoietic cell transplantation.

**Table 1 viruses-15-01286-t001:** Demographic and clinical characteristics of patients with suspected refractory CMV DNAemia.

Patient Characteristics	No. of Patients (%)				*p* Value
	Total Patients	Resistance Variants	VUSs	Without Any Variants	
Number of patients	123 (100.0%)	10 (100.0%)	48 (100.0%)	65 (100.0%)	-
Age (years; range)	44; 5–72	39; 5–67	46; 11–72	44; 8–71	0.531
Male	55 (44.7%)	4 (40.0%)	23 (47.9%)	28 (43.1%)	0.835
Primary disease					
AML	50 (40.7%)	3 (30.0%)	22 (45.8%)	25 (38.5%)	0.567
ALL	24 (19.5%)	3 (30.0%)	11 (22.9%)	10 (15.4%)	0.415
MDS	14 (11.4%)	2 (20.0%)	6 (12.5%)	6 (9.2%)	0.579
NHL	11 (8.9%)	1 (10.0%)	3 (6.3%)	7 (10.8%)	0.702
Aplastic anemia	9 (7.3%)	0	1 (2.1%)	8 (12.3%)	0.077
Others	15 (12.2%)	1 (10.0%)	5 (10.4%)	9 (13.8%)	0.839
HCT source					
Peripheral blood	106 (86.2%)	8 (80.0%)	40 (83.3%)	58 (89.2%)	0.561
Cord blood	17 (13.8%)	2 (20.0%)	8 (16.7%)	7 (10.8%)	-
Related donor HCT	74 (60.2%)	4 (40.0%)	30 (62.5%)	40 (61.5%)	0.395
HLA-matched HCT	62 (50.4%)	6 (60.0%)	26 (54.2%)	30 (46.2%)	0.574
Letermovir prophylaxis	13 (10.6%)	1 (10.0%)	4 (8.3%)	8 (12.3%)	0.792
CMV D/R serostatus					
D+/R+	66 (53.7%)	3 (30.0%)	26 (54.2%)	37 (56.9%)	0.282
D+/R-	4 (3.3%)	0	2 (4.2%)	2 (3.1%)	0.791
D-/R+	26 (21.1%)	3 (30.0%)	10 (20.8%)	13 (20.0%)	0.769
D-/R-	0	0	0	0	-
D?/R+	27 (22.0%)	4 (40.0%)	10 (20.8%)	13 (20.0%)	0.353
Development of refractory CMV DNAemia (days)	70	66.5	72	70	0.942
CMV organ disease					
One	39 (31.7%)	4 (40.0%)	12 (25.0%)	23 (35.4%)	-
Two or more	11 (8.9%)	4 (40.0%)	3 (6.3%)	4 (6.2%)	-
No	73 (59.3%)	2 (20.0%)	33 (68.8%)	38 (58.5%)	0.017
Highest CMV viral load (IU/mL; range)	593,178.4; 910–14,690,000	2281,671.8; 36,080–14,690,000	303,385.7; 3668–2,444,000	543,370.9; 910–10,378,623	0.015 *
Acute GVHD					
None to mild	35 (28.5%)	1 (10.0%)	8 (16.7%)	26 (40.0%)	0.003 **
Moderate to severe	88 (71.5%)	9 (90.0%)	40 (83.3%)	39 (60.0%)	-
One-year survival rate after HCT (%)	52.9%	40.0%	43.8%	61.5%	0.044 **

CMV, cytomegalovirus; VUS, variant of uncertain significance; AML, acute myeloid leukemia; ALL, acute lymphoblastic leukemia; MDS, myelodysplastic syndrome; NHL, non-Hodgkin lymphoma; HCT, hematopoietic cell transplantation; HLA, human leukocyte antigen; D+, donor seropositive; D-, donor seronegative; D?, donor serostatus unknown; R+, recipient seropositive; R-, recipient seronegative; GVHD, graft-versus-host disease. * Comparison between patients with resistance variants and others; ** Comparison between patients with any genetic variants and those without variants.

## Data Availability

The data presented in this study are available from the corresponding author upon reasonable request.

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
