# Peer review of "Genetic Variants Associated with Drug Resistance of Cytomegalovirus in Hematopoietic Cell Transplantation Recipients"

_viruses, 2023, doi:10.3390/v15061286_

Round 1

Reviewer 1 Report

Comments and Suggestions for Authors

The manuscript by Chae et al reports on the analysis of HCMV variants associated with drug resistance after hematopoietic cell transplantation (HCT) and concludes that the identification of drug resistant HCMV in HCT recipients is important for determining appropriate antiviral treatment and for predicting patient outcomes.

I have few problems with the data presented or the interpretation of these data. That said, a valid question is the level of novelty of the findings. As the authors have said in the first paragraph of their discussion, much of their data (overall infection rate among HCT recipients; suspected refractory CMV DNAemia; numbers of resistance variants etc) were all “in line with previous studies”.

The view of the authors that their findings highlight a potential association between virus variants and reduced antiviral efficacy does suggest that variations in virus sequence might be useful for identify patients who need alternative treatment approaches but, again, I am uncertain how novel this is.

Reviewer 2 Report

Comments and Suggestions for Authors

The manuscript by Chae et al describes genetic variants associated with drug resistance of CMV in HCT recipient.  The study is based on a retrospective sample studies of 2271 patients who received HCT between 2016 and 2021.  Direct sequencing reveals a link of genetic variant in UL97 and UL54 region with drug resistance.  Previous reports have also linked genetic variants in this region with drug resistance. The procedures and data analysis are straight forward. Studies identifying genetic variants associated with CMV drug resistant in HCT recipient important in providing antiviral treatment and predict patient outcomes. The following issues need to be addressed to improve the current manuscript to warrant publication.

1.      Association of drug resistance with UL97 and UL54 region have been described previously.  The authors need to explain and describe that the novelty and the uniqueness of their study that warranty publication convincing as new findings. 

2.     The authors need to clearly demonstrate the correlation of genetic variants to drug resistance.  How did they monitor the direct correlation of genetic change to drug resistance?

3.     The authors describe sample collection at different time point.  Is there a time course analysis of the rise of genetic variants?

Comments on the Quality of English Language

Minor editing required

Round 2

Reviewer 2 Report

Comments and Suggestions for Authors

The Authors attempted to explain the concern raised about the novelty of their study.  Although the novelty of their report still remains unclear the information provided could serve of an additional contribution to the already known information.